*Review*

# The functional diversity of protein lysine methylation

Sylvain Lanouette, Vanessa Mongeon, Daniel Figeys & Jean-François Couture[*]

## Abstract

Large-scale characterization of post-translational modifications (PTMs), such as phosphorylation, acetylation and ubiquitination, has highlighted their importance in the regulation of a myriad of signaling events. While high-throughput technologies have tremendously helped cataloguing the proteins modified by these PTMs, the identification of lysine-methylated proteins, a PTM involving the transfer of one, two or three methyl groups to the ε-amine of a lysine side chain, has lagged behind. While the initial findings were focused on the methylation of histone proteins, several studies have recently identified novel non-histone lysine-methylated proteins. This review provides a compilation of all lysine methylation sites reported to date. We also present key examples showing the impact of lysine methylation and discuss the circuitries wired by this important PTM.

**Keywords** lysine demethylation; lysine methylation; networks; proteomics; systems biology

**Mol Syst Biol. (2014) 10: 724**

## Introduction

Covalent post-translational modifications (PTMs) of proteins create an intricate layer of modulation of the proteome. The convergence of high-throughput proteomics efforts with targeted studies of site-specific PTM and protein-modifying enzymes has shed light on the scope of these modifications across a wide variety of organisms. Among the 20 amino acids, lysine is one of the most heavily modified. To this day, lysine residues are known to be covalently modified by acetyl (Choudhary *et al*, 2009; Weinert *et al*, 2011; Henriksen *et al*, 2012), hydroxyl (Van Slyke & Sinex, 1958), glycosyl (Johansen *et al*, 2006), propionyl (Chen *et al*, 2007; Cheng *et al*, 2009), butyryl (Chen *et al*, 2007), crotonyl (Tan *et al*, 2011), ubiquitinyl and ubiquitinyl-like (SUMOylation, ISGylation and NEDDylation) (Hochstrasser, 2009; Kim *et al*, 2011a; Wagner *et al*, 2011), formyl (Wisniewski *et al*, 2008), malonyl (Peng *et al*, 2011), succinyl (Zhang *et al*, 2011b; Park *et al*, 2013; Weinert *et al*, 2013) and methyl (Lan & Shi, 2009; Yang *et al*, 2009b; Egorova *et al*, 2010; Stark *et al*, 2011) groups. Among those modifications, lysine methylation represents a complex and often elusive PTM that has nonetheless the potential to alter the function of the modified protein. This widespread PTM, which involves the transfer of up to three methyl groups to the ε-amine of a lysine residue, has drawn considerable attention in recent years. To this day, lysine methylation has been observed in both nuclear and cytoplasmic proteins and is now considered a prevalent modification in eukaryotes, prokaryotes and archaea (Iwabata *et al*, 2005; Jung *et al*, 2008; Botting *et al*, 2010; Pang *et al*, 2010). Here, we review the range of lysine methylation, its regulation, dynamics and effects.

### Uncovering lysine methylation

Methylation of a lysine residue was first reported in 1959 by Ambler and Rees (1959), in the flagellin protein of *Salmonella typhimurium*. While the origin and the function of the methyllysine residue was a mystery at the time, the observation that histone proteins were also methylated suggested that this PTM is a prevalent modification (Murray, 1964). The subsequent discovery of the methylation of a wide range of proteins (DeLange *et al*, 1969, 1970; Hardy & Perry, 1969; Hardy *et al*, 1970; Ames & Niakido, 1979; L'Italien & Laursen, 1979; Bloxham *et al*, 1981; Motojima & Sakaguchi, 1982; Tong & Elzinga, 1983) confirmed the predominance of this PTM in both prokaryotes and eukaryotes.

In addition, the regulation of EF-Tu methylation by carbon, phosphorus or nitrogen availability (Young *et al*, 1990) and the evolutionarily conserved character of multiple methylation sites identified in ribosomal proteins (Dognin & Wittmann-Liebold, 1980; Amaro & Jerez, 1984; Lhoest *et al*, 1984; Guérin *et al*, 1989) hinted that lysine methylation could serve important biological functions. This was confirmed by the report that methylation of calmodulin K115 (Watterson *et al*, 1980; Marshak *et al*, 1984; Lukas *et al*, 1985) lowers its capacity to stimulate NAD kinase activity (Roberts *et al*, 1986). Methylation of calmodulin does not, however, prevent the activation of other calmodulin targets (Molla *et al*, 1981; Roberts *et al*, 1986). These findings showed that lysine methylation modulates the function of a protein and demonstrated that this PTM has the ability to affect only a subset of activity of the methylated substrate.

Interest in lysine methylation intensified following the observation that the methylation of lysine 9 on histone H3 leads to the

Ottawa Institute of Systems Biology, Department of Biochemistry, Microbiology and Immunology, University of Ottawa, Ottawa, Canada
*Corresponding author. Tel: +1 613 562 5800 8854; Fax: +1 613 562 5655; E-mail: jean-francois.couture@uottawa.ca

recruitment of HP1 (Swi6 in *S. pombe*) to chromatin (Bannister *et al*, 2001; Lachner *et al*, 2001) and consequently promotes heterochromatin formation. This effect suggested that the widespread modification of histone proteins by methylation could lead to dramatic effects on gene expression.

## Protein lysine methyltransferases

Two groups of enzymes, both using S-adenosyl-L-methionine (SAM) as a methyl donor, catalyze the addition of a methyl group to the ε-amine group of a lysine side chain (Schubert *et al*, 2003). The first type of protein lysine methyltransferase regroups the enzymes containing a catalytic SET domain (class V methyltransferases). The SET domain, named after SU(var), Enhancer of Zeste and Trithorax, the three first identified proteins harboring this domain in *Drosophila* (Tschiersch *et al*, 1994), is characterized by three regions folded into a mainly β-sheet knot-like structure that forms the active site consisting of the four conserved motifs GXG, YXG, NHXCXPN and ELXFDY (Dillon *et al*, 2005; Qian & Zhou, 2006; Cheng & Zhang, 2007). Binding of SAM and the substrate takes place on each side of a methyl-transfer channel formed by this knot-like structure. It is suggested that a catalytic tyrosine resting in this channel is important for the methyl transfer from SAM to the lysine ε-amine (Min *et al*, 2002; Trievel *et al*, 2002, 2003; Wilson *et al*, 2002; Kwon *et al*, 2003; Manzur *et al*, 2003; Xiao *et al*, 2003; Couture *et al*, 2006). A network of aromatic residues and hydrogen bonds in this channel limits the possible orientations of the lysine substrate (Couture *et al*, 2008), controlling the ability of SET domain proteins to transfer a specific number of methyl groups to a substrate.

Based on sequence similarities and domain organization, the SET-domain-containing proteins can be broadly divided in seven families (Dillon *et al*, 2005): SUV3/9, SET1, SET2, SMYD, EZ, SUV4-20 and RIZ. Members of the SUV3/9 (G9a (Rathert *et al*, 2008b), GLP (Chang *et al*, 2011), SETDB1 (Van Duyne *et al*, 2008)), SET1 (SET1 (Zhang *et al*, 2005)), SET2 (NSD1 (Lu *et al*, 2010)), SMYD (SMYD2 (Huang *et al*, 2006), SMYD3 (Kunizaki *et al*, 2007)) and EZ (EZH2 (He *et al*, 2012a)) families methylate both histone and non-histone substrates (Supplementary Table S1 and Fig 1), while substrates reported to this day for the SUV4-20 and RIZ families are limited to histone proteins (Yang *et al*, 2008; Pinheiro *et al*, 2012). Outside of these seven families, SET7/9 and SET8 are also reported to methylate a substantial number of proteins (Table 1, Supplementary Table S1 and Fig 1).

The second class of PKMTs, the seven β-strand methyltransferases (class I methyltransferases), belongs to an extended superfamily of methyltransferases found throughout eukaryotes, prokaryotes and archaea. Members of this family methylate DNA, RNA or amino acids such as arginine, glutamine, aspartate and histidine (Martin & McMillan, 2002; Schubert *et al*, 2003). They are named after its Rossmann fold built around a central β-sheet structure, which includes the conserved, catalytic motifs hhXhD/E, XDAX and PXVN/DXXLIXL (h=hydrophobic residue) that allow the association of SAM and the protein substrate.

Across all three domains of life, a number of class I methyltransferases are reported to methylate lysine residues in proteins (Table 1 and Supplementary Table S1). The bacterial methyltransferases PrmA and PrmB methylate the ribosomal units L11 (Cameron *et al*, 2004) and L3 (Colson *et al*, 1979), respectively (Supplementary Table S1). In *S. cerevisiae*, Rkm5 methylates the ribosomal protein L1ab (Webb *et al*, 2011) and See1 methylates the elongation factor EF1-α on K316 (Lipson *et al*, 2010) (Supplementary Table S1; Fig 1). Recently, VCP-KMT, a newly identified class I methyltransferase, was shown to methylate the membrane protein VCP (Kernstock *et al*, 2012). Class I methyltransferases are also able to methylate histones, as Dot1 homologs trimethylate K79 of histone H3 (Nguyen & Zhang, 2011). In crenarchaea, the methyltransferase aKMT, a broad specificity class I lysine methyltransferase, was shown to methylate the DNA-binding protein Cren7 (Chu *et al*, 2012) (Table 1; Fig 1).

## Detection of lysine methylation

Systematic high-throughput studies helped uncover the global implication of PTMs such as phosphorylation (Ptacek *et al*, 2005; Sopko & Andrews, 2008) and acetylation (Choudhary *et al*, 2009; Weinert *et al*, 2011; Henriksen *et al*, 2012) in different cellular processes. If the terms "phosphorylome" and "acetylome" can now properly be applied to our understanding of those modifications, an exhaustive description of the lysine methylome and the biological functions it regulates has yet to be produced. The challenges still associated with the detection of lysine methylation impede research on this PTM. The small molecular weight of a methyl group relative to other PTMs and the lack of a charge difference between methylated and unmethylated lysine residues leave few options for the detection of methylated lysine residues via direct physicochemical methods.

### Targeted discovery of lysine methylation

Given the challenges associated with its detection, the identification of lysine methylation has long relied on the targeted identification of single sites by amino acid sequencing, radio-labelled assays or immunoblotting. Some of the earliest reports of lysine methylation were provided by Edman sequencing (Bloxham *et al*, 1981; Tong & Elzinga, 1983; Schaefer *et al*, 1987; Ammendola *et al*, 1992). This method is reliable and precise enough to detect methyllysine (Fig 2A). However, Edman sequencing is time-consuming and necessitates large amounts of the target proteins, making it inapplicable to high-throughput approaches. Introduction of radioactively labelled methyl donors either in culture media or lysate (Fig 2A,B) has also been used to detect methylated proteins in model systems, together with 2D SDS–PAGE or liquid chromatography (Dognin & Wittmann-Liebold, 1980; Wang *et al*, 1982, 1992; Wang & Lazarides, 1984). The use of radioactive material on this scale is however cumbersome and does not allow the identification of specific methylation sites. It also does not indicate what type of residue is labelled, as arginine, histidine, aspartate and glutamate residues as well as the amino terminus of proteins can be the targets of S-adenosyl-L-methionine-dependent methyltransferase (Stock *et al*, 1987; Webb *et al*, 2010; Petrossian & Clarke, 2011). More recent studies have made use of immunoblotting to explore potential methylation sites on proteins (Iwabata *et al*, 2005). However, pan-methyllysine antibodies suffer from a low level of specificity, sensitivity and low reproducibility between suppliers and lots available. As for generic radioactive methylation assays, immunoblotting with pan-methyllysine antibodies does not allow the determination of the

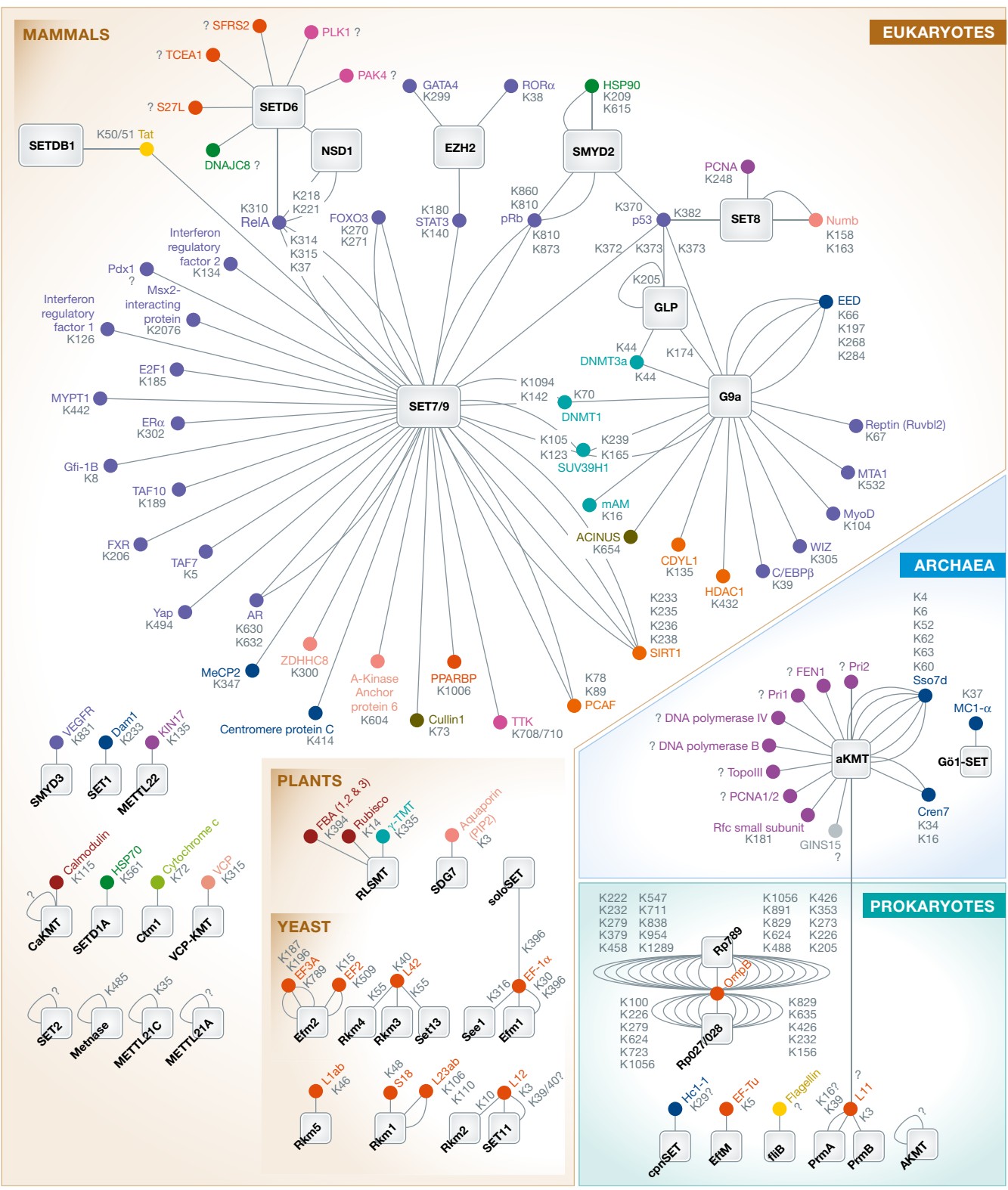

**Figure 1.  PKMT–substrate association maps suggest that lysine methylation is found in complex regulatory networks.**
Each PKMT or substrate node of the methylation networks is color-coded according to its functional classification (see Supplementary Table S1). In Eukarya, 34 PKMTs methylate > 65 substrates other than histones. SET7/9 is by far the most promiscuous PKMT targeting close to half of eukaryotic substrates reported to this day. In contrast to eukaryotes, only 8 unique PKMTs have been identified in prokaryotes and 2 in Archaea. Those interactions, together with the 1,018 methylation sites listed in Supplementary Table S1, demonstrate the complexity of this modification and its regulatory potential for the proteome.

**Table 1. Lysine methylation is a prominent post-translational modification**

Transcription

| Protein | Uniprot ID | Lysine | State | Organism | KMT | KDM | Evidence | Effects | References |
|---|---|---|---|---|---|---|---|---|---|
| p53 | P04637 | 370 | 1Me | Hs | SMYD2 | LSD1 | viv, vit; AB, RD, MS | Represses p53 activity and prevents methylation of K372 | Huang (2007) Nature |
| | | | 2Me | Hs | ? | LSD1 | | Prevents 53BP1 binding (represses p53 activity) | Huang (2006) Nature (LSD1 : Huang et al (2007) Nature) |
| | | | | | | | | Recruits PHF20 with K382Me2 and inhibits p53Ub after DNA damage | Cui (2012) NSMB |
| | | 372 | 1Me | Hs | 9/9 | LSD1 | viv, vit; AB, RD | Stabilizes p53 | Chuikov (2004) Nature (LSD1 : Huang et al (2007) Nature) |
| | | | | | | | | Inhibits methylation at K370 by SMYD2 | Huang et al (2007) Nature |
| | | | | | | | | Promotes acetylation of p53 (K373, K382) | Ivanov (2007) Mol Cell. Bio. |
| | | | | | | | | Inhibited by HPV E6 and protects p53 from E6-mediated degradation | Hsu et al (2012) Oncogene |
| | | | | | | | | In contrast, deletion of SET7/9 in mice does not impair p53 function, anti-oncogenic activity, transcriptional activity, or its acetylation | Campaner (2011) Mol. Cell, Lehnertz (2011) Mol. Cell |
| | | 373 | 2Me | Hs | G9a (Glp) | ? | viv, vit; AB, RD | Inhibits apoptotic activity | Huang (2010) JBC |
| | | | | | | | | Stimulated by recruitment of MDM2 and correlates with H3K9Me3 at p21 promoter | Chen (2010) EMBO J. |
| | | 382 | 1Me | Hs | SET8 | ? | viv, vit; AB, MS | Suppresses transcriptional activation of highly responsive target genes | Shi (2007) Mol. Cell |
| | | 382 | 2Me | Hs | ? | ? | viv; AB, MS | Recruits L3MBTL1 through MBT repeats | West (2010) JBC |
| | | | | | | | | Corelates with DNA damage and facilitates 53BP association | Kachirskaia et al (2008) JBC |
| | | 386 | 1/2Me | Hs | ? | ? | viv; MS | Recruits PHF20 with K382Me2 and inhibits p53Ub after DNA damage | Cui et al (2012) NSMB |
| | | | | | | | | ? | Kachirskaia (2008) JBC |

**Table 1** (continued)

| Protein | Uniprot ID | Lysine | State | Organism | KMT | KDM | Evidence | Effects | References |
|---|---|---|---|---|---|---|---|---|---|
| pRb | P06400 | 810 | 1Me | Hs | SET7/9 | ? | viv, vit; RD, MS | Elicited by DNA damage and cell cycle arrest, impairs Cdk binding and Pi of Rb | Carr et al (2011) EMBO J |
| | | 810 | 1Me | Hs | SMYD2 | ? | viv, wit; AB, RD, MS | Enhances Pi of S807/S811 and promotes cell cycle progression through E2F activity | Cho (2012) Neoplasia |
| | | 860 | 1Me | Hs | SMYD2 | ? | viv; AB, RD, MS | L3MBTL1 binding (repressor of target genes) | Saddic (2010) JBC |
| | | 873 | 1Me | Hs | SET7/9 | ? | viv; AB, RD | Required for cell cycle arrest and transcriptional repression and recruits HP1 to pRb | Munro (2010) Oncogene |
| E2F1 | Q01094 | 185 | 1Me | Hs | SET7/9 | LSD1 | viv, vit; RD | Decreases stability (increases ubiquitination) and impairs PCAF-Ac and CHK2/ATM-Pi (activating modifications) | Kontaki (2010) Mol. Cell |
| | | | | | | | | Prevents NEDDylation of E2F1, protecting its activity | Loftus et al (2012) EMBO rep. |
| | | | | | | | | Inhibited by TMCG/DIPY which reduces RASSF1A expression | Montenegro et al (2012) PLoS One |
| | | | | | | | | Levels correlate with DNA damage and increases DNA binding | Xie (2011) J. Recept. Signal Transduct. Res. |
| NF-kB (p65, RelA) | Q04206 | 37 | 1Me | Hs | SET7/9 | ? | viv, vit; AB, RD, MS | Regulates p65 promoter binding, necessary for certain target genes | Ea (2009) PNAS |
| | | 218/221 | 1Me/2Me | Hs | NSD1 | FBXL11 | viv; AB, MS | Activation of NF-kB; K221 recruits PHF20 that prevents PP2A recruitment and protects Pi and Ac of p65 | Lu (2010) PNAS, PHF20: Zhang et al (2013) Nature Comm. |
| | | 310 | 1Me | Hs | SETD6 | ? | viv, vit; AB, RD, MS | Recognized by ankyrin repeat domain of GLP that is recruited to RelA target genes and upregulates H3K9Me2 levels and downregulates their expression (GLP association inhibited in turn by S311Pi by PKC-ζ, which then allows expression of target genes) | Levy et al (2011) Nat. Immunol., Chang (2011) Nucl. Ac. Res. |

**Table 1** (continued)

| Protein | Uniprot ID | Lysine | State | Organism | KMT | KDM | Evidence | Effects | References |
|---|---|---|---|---|---|---|---|---|---|
| | | 314/315 | 1Me | Hs | SET7/9 | ? | viv, vit; AB*, RD, MS | Stimulated by TNF-α; induces degradation of promoter-associated RelA (proteasome); stimulated by berberine and leads to ROS production | Yang (2009a,b) EMBO J. (Berberine: Hu et al (2013) Acta Pharm. Sin.) |
| | | | | | | | viv, vit; AB, RD | Inhibited by K310Ac (which is opposed by SIRT1) | Yang (2010a,b) Mol. Cell. Biol. |
| TAF10 | Q12962 | 189 | 1Me | Hs | SET7/9 | ? | viv, vit; AB*, RD | Potentiates transcription of certain TAF10-dependent genes | Kouskouti (2004) Mol. Cell |
| GATA4 | P43694, Q08369 | 299 | 1Me | Hs, Mm | EZH2 (PRC2) | ? | viv, vit; MS, AB*, RD | Requires SUZ12 and EED, occurs in fetal hearts, prevents GATA4 C-terminal AcK by p300, limits GATA4-mediated recruitment of p300 to chromatin which represses the expression of these target genes | He (2012) Genes Dev. |
| Reptin (Ruvbl2) | Q9Y230 | 67 | 1Me | Hs | G9a | ? | viv; MS, AB* | Negatively regulates hypoxia-responsive genes | Lee (2010) Mol. Cell |
| C/EBPβ | Q05826, P28033, P21272 | 39 | ? | Mm, Rn | G9a | ? | viv, vit; RD | Inhibition of transactivation potential | Pless (2008) JBC |
| ARID5B | Q14865 | 336 | 2Me | Hs | ? | PHF2 | viv; MS, AB | Demethylation of ARID5B necessary for binding to target promoters | Baba (2011) Nat. Cell. Biol. |
| ERα | P03372 | 302 | 1Me | Hs | SET7/9 | ? | viv, vit; MS, RD, AB | Recruitment of ER to target genes and transactivation | Subramanian et al (2008) Mol. Cell |
| | | 472 | 3Me | Hs | ? | ? | viv; MS | Could be AcK | Atsriku et al (2009) MCP |
| AR | P10275 | 630 | 1Me | Hs | SET7/9 | ? | viv, vit; RD, AB* | Enhances AR transactivation through interdomain (N-C) interaction | Ko et al (2011) Mol. Endocr. |
| | | 632 | 1Me | Hs | SET7/9 | ? | viv, vit; RD, AB* | Enhances transcriptional activity and recruitment to target genes, site disputed | Gaughan et al (2011) Nucl. Acid Res. (disputed: Ko et al (2011) Mol. Endocr.) |
| Chromatin/chromosomal regulation | | | | | | | | | |
| Dam1 | P53267 | 233 | 2Me | Sc | SET1 | ? | viv; AB | Tunes levels of Pi for S232, S234, and S235 by Ipl1; important for proper chromosome segregation | Zhang (2005) Cell |
| | | | | | | | | Occurs at kinetochore and necessitates Paf1 | Latham (2011) Cell |

**Table 1** (continued)

| Protein | Uniprot ID | Lysine | State | Organism | KMT | KDM | Evidence | Effects | References |
|---|---|---|---|---|---|---|---|---|---|
| | | | | | | | | independently of transcriptional elongation | |
| | | | | | | | | Requires H2BK123Ub; requires Rad6 & Bre1; Ubp8 dowregulates levels | Latham *et al* (2011) Cell |
| MC1-α | Q8PY15 | 37 | ? | *M. mazei* | Gö1-SET | ? | vit; RD | ? | Manzur (2005) FEBS letters |
| Cren7 | Q97ZE3,C3N5A6 | 16 | 1Me/2Me | *S. solfataricus S. islandicus* | aKMT | ? | viv, vit; RD, MS | ? | Guo *et al* (2008) Nucl. Ac. Res.; aKMT: Chu (2012) J. Bact. |
| | | 34 | 1Me | *S. solfataricus S. islandicus* | aKMT | ? | viv, vit; RD, MS | ? | Guo *et al* (2008) Nucl. Ac. Res.; aKMT: Chu *et al* (2012) J. Bact. |
| | | 31, 37, 42? | ?Me | *S. solfataricus* | ? | ? | viv; MS | ? | Guo *et al* (2008) Nucl. Ac. Res. |
| Protein synthesis | | | | | | | | | |
| EF-Tu | P09591 | 5 | 3Me | *P. aeruginosa* | EftM | ? | viv; MS | Mimics platelet-activating factor to mediate interaction with PAF receptor and allows bacterial invasion in pneumonia | Barbier (2013) Pneumonia |
| | P0CE47, P02991, P0A1H5, P33166, Q65PA9 | 56 | 1Me/2Me | *E. coli, E. gracillus, S. typhimurium, B. subtilis, B. licheniformis* | ? | ? | viv; RD | Affects bound tRNA conformation, lowers GTPase activity (2Me in stationary phase) and hypermethylation controlled by the availability of carbon, nitrogen, and phosphate sources in external medium; induces dissociation of EF-Tu from membranes | L'Italien (1979) FEBS Lett. (role: Van Noort (1986) Eur. J. Biochem, Young (1991) J. Bacteriol., Toledo & Jerez (1990)) |
| RL1ab | P0CX43, P0CX44 | 46 | 1Me | Sc | Rkm5 | ? | viv, vit; MS, RD | No effect versus protein synthesis inhibitors | Webb (2011) JBC |
| RL12 | P0CX53, P0CX54, P30050, O75000, Q9W1B9 | 3 | 3Me | Sp, Hs, Dm | SET11 | ? | viv, vit; MS, RD | "Growth defect" if SET11 overexpressed; recruits Corto chromodomain to *Drosophila* nucleus which recruits RNAPol III to chromatin and activates transcription | Sadaie *et al* (2008) JBC; Corto: Coleno-Costes (2012) PLOS genet. |
| | | 39, 40? | 3Me? | Sp | SET11 | ? | | | Sadaie *et al* (2008) JBC |

**Table 1** (continued)

| Protein | Uniprot ID | Lysine | State | Organism | KMT | KDM | Evidence | Effects | References |
|---|---|---|---|---|---|---|---|---|---|
| | | | | | | | viv, vit; MS, RD | ? – "Growth defect" if SET11 overexpressed | |
| RL23ab | A6ZKL6 | 106 | 2Me | Sc | Rkm1 | ? | viv; RD | No effect on RNA binding, may affect Rpl23ab position in the large subunit | Porras-Yakushi et al (2007) JBC |
| | | 110 | 2Me | Sc | Rkm1 | ? | viv; RD | No effect on RNA binding, may affect Rpl23ab position in the large subunit | Porras-Yakushi et al (2007) JBC |
| RL42 | P0CX27, P0CX28, Q9UTI8 | 40 | 1Me | Sc | Rkm3 | ? | viv; MS | ? | Webb et al (2008) JBC, Couttas et al (2012) Proteomics |
| | | 55 | 1Me | Sc, Sp | SET13 (Sc:Rkm4) | ? | viv; MS | Stress protection, survival in stationary phase, cycloheximide protection | Shirai (2010) JBC (Sc: Webb et al (2008) JBC) |
| **Methyltransferases/Demethylases** | | | | | | | | | |
| DNMT1 | P26358, P13864 | 70 | 2Me | Hs | G9a | ? | vit; RD | ? | Rathert (2008a) Rathert (2008b)Nat. Chem. Biol. |
| | | 142 | 1Me | Hs, Mm | SET7/9 | LSD1 | viv, vit; MS, AB, RD | Susceptibilize DNMT1 to proteasome degradation; inhibited by S143Pi by AKT1 | Esteve (2009) PNAS (LSD1 : Wang (2009) Nat Genet.; PiS: Esteve (2011)NSMB)) |
| | | | | | | | | Cyclophosphamide increases levels of K142Me (increases LSD1) | Zhang (2011a) Zhang (2011b) Chem. Res. Toxicol. |
| | | 1094 | ? | Mm | SET7/9 | LSD1 | viv, vit; RD | Reduces stability of DNMT1, decreases global levels of DNA methylation | Wang et al (2009) Nature Genet. |
| DNMT3a | Q9Y6K1, O88508 | 44 | 2Me | Hs, Mm | G9a, GLP | ? | viv,vit; MS, AB, RD | Recruits MPP8 chromodomain to DNMT3a; possible role in G9a/GLP/ DNMT3a/ MPP8 complex formation | Chang et al (2011) Nature Comm. |
| G9a | Q96KQ7 | 94 | 2/3Me | Hs | ? | ? | viv, vit; MS, AB | ? | Sampath et al (2007) Mol. Cell |
| | | 114 | 3Me | Hs | ? | ? | viv; MS* | ? | Bremang (2013) Mol. BioSyst. |
| | | 165 | 2/3Me | Hs | G9a | ? | viv, vit; MS, AB | Recruits HP1 (reversed by T166Pi); recruits Cbx3 | Sampath et al (2007) Mol. Cell, Ruan et al (2012) Mol. Cell |
| | | 239 | 3Me | Hs | G9a | ? | vit; MS, RD | Colocalization of HP1 with G9a | Chin et al (2007) Nucl. Acids Res. |

**Table 1** (continued)

| Protein | Uniprot ID | Lysine | State | Organism | KMT | KDM | Evidence | Effects | References |
|---|---|---|---|---|---|---|---|---|---|
| GLP | Q9H9B1 | 122 | 3Me | Hs | ? | ? | viv; MS* | ? | Bremang et al (2013) Mol. BioSyst. |
|  |  | 174 | ? | Hs | G9a | ? | vit; RD | *In vitro* evidence only | Chin et al (2007) Nucl. Acids Res. |
|  |  | 205 | ? | Hs | GLP | ? | viv, vit; MS, RD, AB | Recruits MPP8 and GLP; possible role in G9a/GLP/DNMT3a/MPP8 complex formation | Chang et al (2011) Nature Comm. |
| Chaperones |  |  |  |  |  |  |  |  |  |
| HSP90 | P07900 | 615 | 1Me | Hs, Mm, Dr | SMYD2 | LSD1 | viv, vit; MS, RD, AB | Correlates with association of a SMYD2/HSP90 complex to titin and correct myofilament organization | Abu-Fahra (2011) J. Mol. Cell Biol., Donlin (2012) Genes & Dev, Voelkel (2013) BBA |
| HSP70 | P08107 | 561 | 2Me | Hs | SETD1A | ? | viv, vit; MS, AB | Promotes association with AURKB which enhances its activity; enhances cancer cell growth | Cho et al (2012a, 2012b) Nature Comm., Cloutier et al (2013) PloS Genet. |
| Metabolism |  |  |  |  |  |  |  |  |  |
| Calmodulin | P62152 | 94 | 1Me/2Me | Dm | ? | ? | viv, MS | Eye specific | Takemori et al (2007) Proteomics |
|  | P62158, P62161, P06787, P07463 | 115 | 3Me | Hs, Rn, Oa, Nt, Ps, Sc, Sp, *P. tetraurelia* | CaKMT | ? | viv, vit; ED, RD, MS | Reduces NAD kinase activation; reduces *in vitro* $T_m$ of linker region, not required for myosin light chain activation; role in stem internode growth, seed production and seed and pollen viability; for mammals, no effect on cell growth, proliferation or calmodulin stability; necessitate chaperoning of CAKMT by middle domain of HSP90 | Watterson 1980 (JBC) (role: Roberts et al (1986) JBC; myosin: Molla (1981) JBC, Roberts et al (1992) PNAS CAKMT: Sitaramayya et al (1980) JBC, Oh & Roberts (1990) Plant Physiol, Han et al (1993) Biochemistry, Pech1994 BBA, Magnani et al (2010) Nat. Comm.; Tm: Magnani et al (2012) Protein: Expr. and Pur. Mammals; Panina et al (2012) JBC ; HSP90 : Magen et al (2012) PLoS One), Bremang et al (2013) Mol. BioSyst. |

**Table 1** (continued)

| Protein | Uniprot ID | Lysine | State | Organism | KMT | KDM | Evidence | Effects | References |
|---|---|---|---|---|---|---|---|---|---|
| Rubisco | P11383, P00876, P04717, P27064 | 14 | 3Me | T. aestivum, Nt, P. sativum, Solanaceae, Cucurbitaceae | RLSMT | ? | viv, vit; ED, RD, MS | ? - Not methylated in Arabidopsis | Houtz et al (1989) PNAS, Houtz et al (1992) Plant Physiol. (RLSMT: Houtz et al (1991) Plant Physiol. At: Mininno (2012) JBC) |
| β-glycosidase | P22498 | 116/135? | 1Me/2Me? | S. solfataricus | ? | ? | viv; MS | Enhances thermal stability | Febbraio et al (2004) JBC |
| | | 272 | 2Me | S. solfataricus | ? | ? | viv; MS | Enhances thermal stability | Febbraio et al (2004) JBC |
| | | 311/322? | 1Me/2Me? | S. solfataricus | ? | ? | viv; MS | Enhances thermal stability | Febbraio (2004) JBC |
| Citrate synthase | P00889 | 368 | 3Me | Ss | ? | ? | viv; ED | No effect on catalysis | Bloxham (1981) PNAS (no effect: Evans et al (1988) BBRC) |
| **Electron transfer & oxidative stress** | | | | | | | | | |
| Cytochrome c | P00044, P00068, P62898, P00048, P00043, P00041 | | 3Me | Rn, Sc, Nc, T. aestivum, N. crassa, H. anomala, D. kloeckeri, C. krusei | Ctm1 | ? | viv, vit; ED, RD | Blocks cytochrome c apoptotic activity; minor role in transfer to mitochondria in yeast; absent from most higher mammals, vertebrates | DeLange (1969) JBC, DeLange (1970) JBC, Sugeno et al (1971) J. Biochem. Brown et al (1973) Biochem. J. (Ctm1p: Polevoda et al (2000) JBC; roles: Kluck et al (2000) JBC) |
| **Viral proteins** | | | | | | | | | |
| Tat | P04610 | 50/51 | 3Me? | HIV-1 | SETDB1 | ? | vit; RD | Inhibits LTR transactivation | Van Duyne (2008) Retrovirology |
| | | 51 | 1Me | HIV-1 | SET7/9 | LSD1 | viv, vit; RD, AB, MS | Enhances HIV transcription, inhibited by K50Ac by p300 but demethylation LSD1 independent of K50Ac; LTR transactivation by LSD1 demethylation | Pagans et al (2010) Cell Host Microbe (LSD1 & Ac interaction : Sakane (2011) PloS Pathog) |
| VP1 | A8Y983 | 5? | 3Me | polyomavirus | ? | ? | viv; RD | ? | Burton & Consigli (1996) Virus Res. |
| **Membrane proteins** | | | | | | | | | |
| VCP | P55072 | 315 | 3Me | Hs | VCP-KMT | ? | viv, vit; RD, MS | Methylated prior to hexamer assembly, does not affect ATPase activity (contested: also observed to lower VCP ATPase activity) | Kernstock (2012) Nature Comm., Lower ATPase activity: Cloutier et al (2013) PloS Genet, Bremang et al (2013) Mol. BioSyst. |

**Table 1** (continued)

| Protein | Uniprot ID | Lysine | State | Organism | KMT | KDM | Evidence | Effects | References |
|---|---|---|---|---|---|---|---|---|---|
| OmpB | Q53020, P96989 | see Supplementary Table S1 | | *R. prowazekii*, *R. typhi* | Rp789, Rp027/028 | ? | viv, vit; RD, MS, AB* | Virulence factor | Chao (2004) BBA, Abeykoon et al (2012) J. Bact. |
| HBHA | A1KFU9, P0A5P6, Q3I5Q7 | 162–195 | 1Me/2Me | *M. bovis*, Mtb, *M. smegmatis* | ? | ? | viv, vit; RD, MS | Possible role in resistance to proteolysis; important for T-cell antigenicity and protective immunity to Mtb infection (only for aerosol infection), non-active TB patients have a stronger response to Me form; does not affect heparin binding | Pethe et al (2002) PNAS, Biet (2007) Micr. & Infect., (antigenicity: Temmerman (2004) Nat. Med., aerosol: Guerrero (2011) Clin. Dev. Immunol., immun response: Delogu (2011) PLoS One) |
| LBP | | ? | ? | *M. smegmatis*, *M. leprae* | ? | ? | viv; MS | Possible role in resistance to proteolysis; does not affect heparin/laminin binding | Pethe (2002) PNAS, (leprae: Soares de Lima (2005) Microbes & Infect.) |

Evidence: viv: *in vivo*, vit: *in vitro*; AB: specific antibody, AB* pan–methyllysine antibody, MS: mass spectrometry, MS*: high-throughput mass spectrometry, ED: Edman degradation, RD, radioactive assay.

methylation site. Antibodies raised against a specific methylation site have however been invaluable in the identification and *in vivo* confirmation of methylated proteins (Fig 2A and Supplementary Table S1).

### High-throughput discovery of lysine methylation

Mass spectrometry is the current method of choice to detect PTMs. This technique is sensitive and reproducible: it can detect the 14 Da shift in the mass of a given peptide corresponding to methyl group and is also capable to determine the residues being methylated (Fig 2B). Its use has nonetheless been impaired by the low abundance, *in vivo*, of methylated sites relatively to their non-methylated counterpart. In addition, the small mass difference between a tri-methylated and an acetylated peptide (42.05 Da versus 42.01 Da) cannot be separated using low-resolution mass spectrometers. Fortunately, the precision of recent instruments such as Orbitrap and triple TOF simplifies their respective identification (Huq *et al*, 2009; Chu *et al*, 2012). Previous proteome-scale studies of acetylation in human cells have used pan-acetyllysine antibodies to enrich acetylated proteins prior to mass spectrometry analysis (Choudhary *et al*, 2009). Low specificity and sensitivity of previously available pan-methyllysine antibodies have limited the use of this approach. Recently, a cocktail of antibodies was developed to enrich methylated peptides (Guo *et al*, 2014) and has successfully yielded a significant number of novel methylation sites. This novel approach identified 165 sites across a wide variety of sequences in histones, elongation factors and chaperone proteins in HCT116 cells. In addition, metabolic labeling methods, such as heavy methyl SILAC (Ong *et al*, 2004), are being developed and have been applied to the *de novo*, high-throughput discovery of chromatin-specific methylation sites (Bremang *et al*, 2013).

Recently, a new approach for the detection of methylation was reported, based on known methyllysine-binding protein domains in lieu of a classic antibody fold (Fig 2B). Liu *et al* used the HP1 β chromodomain as bait against cell extracts and systematic peptide arrays to identify a methyllysine-dependent interactome for the protein (Liu *et al*, 2013). This led to the discovery of 29 new methylated proteins and demonstrated a role of HP1 β in DNA damage response, driven by its interaction with methylated DNA-PKc. Moore *et al*, (2013) also made use of methyl-binding domains by engineering a generic methyl probe from the L3MBTL1 fold. This construct was then used to identify new targets for the PKMTs G9a and GLP directly from cell extracts, utilizing SILAC and specific PKMTs inhibitors.

### Prediction-based discovery of lysine methylation

As an alternative approach to high-throughput technologies, other research groups decided to focus on the determination of substrate recognition by PKMTs. A library of peptides spanning the sequence recognized by a PKMT and bearing targeted or systematic mutations is assayed for methylation optima. These, often together with structural studies, allow for the elucidation of the PKMT specificity and the prediction of new substrates. The approach has so far been applied to G9a (Rathert *et al*, 2008a), SETD6 (Levy *et al*, 2011b), SET7/9 (Couture *et al*, 2008; Dhayalan *et al*, 2011) and SET8 (Kudi-thipudi *et al*, 2012). More specifically, the methyltransferase activity of SET7/9 toward TAF7 (Couture *et al*, 2008), TAF10 (Kouskouti *et al*, 2004) and E2F1 (Kontaki & Talianidis, 2010; Xie *et al*, 2011)

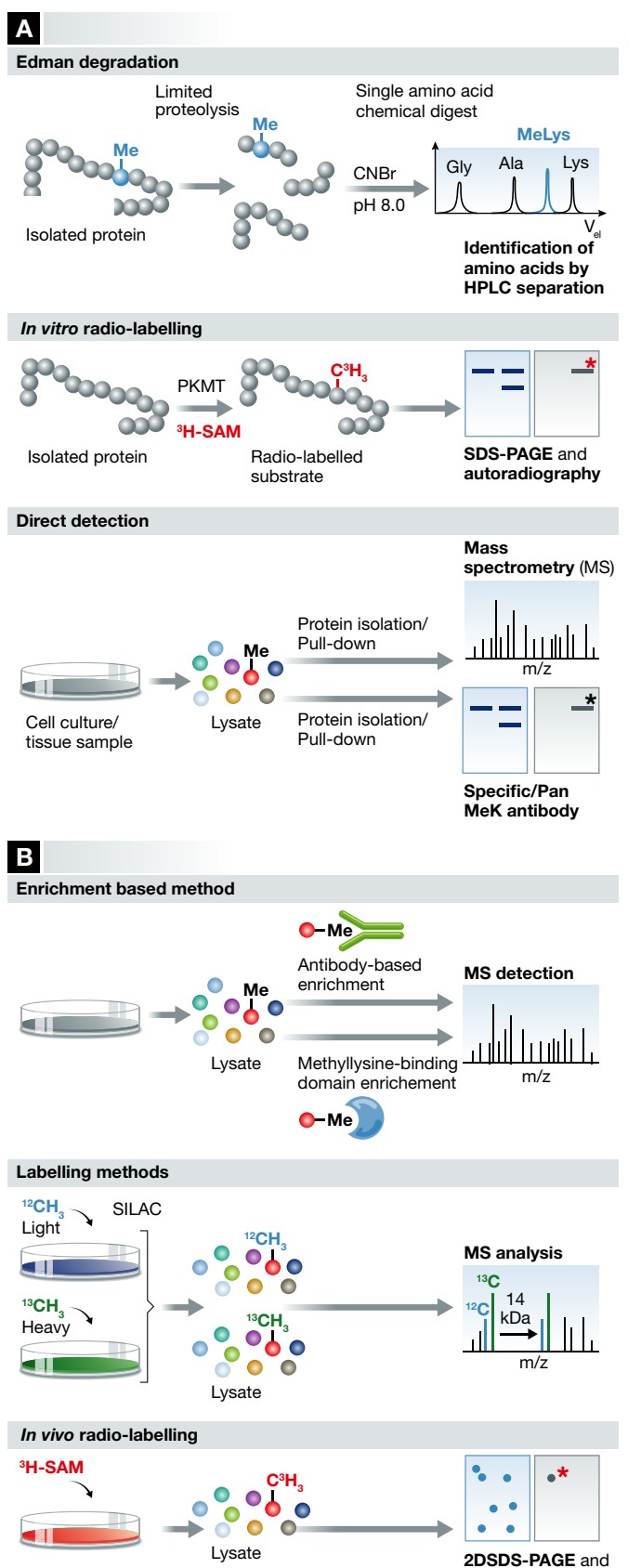

**Figure 2. Detection of lysine methylation.**
(A) Most common experimental approaches in target-specific detection of lysine methylation. Edman degradation and direct detection either by mass spectrometry or by immunoblotting allows for the analysis of *in vivo* samples. *In vitro* radiolabeling is commonly used to confirm the PKMT associated to a given site. (B) Recent high-throughput approaches enabled large-scale identification of methyl-lysine proteins. Methylated peptides or proteins can be enriched, either by pan-methyllysine antibodies or methyl-binding protein domains. Alternately, proteins can be specifically labeled (isotopically, radioactively) to allow an easier identification of methylated peptides.

was first predicted on the basis of methylation assays performed on a small library of peptides (Couture *et al*, 2008). To date, the majority of methylation sites reported for SET7/9 are included within the motif [R/K]-[S/T/A]-K*-[D/K/N/Q] inferred from these assays (Supplementary Table S1). Moreover, a recent study expanded the range of SET7/9 putative substrates (Dhayalan *et al*, 2011). The broader motif identified in this study, [G/R/H/K/P/S/T]-[K/R]-[S/K/Y/A/R/T/P/N]-K*, suggests that SET7/9 may have a more relaxed specificity than previously assessed (Dhayalan *et al*, 2011). An extensive peptide array based on a 21-residue peptide encompassing the N-terminus of histone H3 was also used to characterize the sequence recognized by the methyltransferase G9a (Rathert *et al*, 2008a). The team found that G9a recognizes the motif [N/T/GS]-[G/C/S]-[R]-K*-[T/G/Q/S/V/M/A]-[F/V/I/L/A], where K* is the methylated lysine (Rathert *et al*, 2008a). Among the candidates including this motif, CDYL, WIZ, ACINUS, DNMT1, HDAC1 and Kruppel were shown to be methylated both *in vitro* and *in vivo* by G9a. Furthermore, methylated peptides of the CDYL and WIZ target sequences were found to bind HP1 β chromodomain, demonstrating that methyllysine effectors can recognize those sites. While peptide arrays have proven useful in the identification of protein substrates, this approach may not be applicable to all PKMTs. For example, identification of a motif for SET8 based on a peptide array designed from the tail of histone H4 failed to provide new substrates for this PKMT, demonstrating that peptide substrates may lack important structural determinants required for substrate recognition and catalysis (Kudithipudi *et al*, 2012). In a variation on this approach, full-length protein arrays regrouping over 9000 candidate substrates were used to determine the motif recognized by the methyltransferase SETD6, only known at the time to methylate RelA. A total of 154 total putative targets were predicted. Of these, six substrates were confirmed *in vitro,* and of these, PLK1 and PAK4 were found to be methylated in HEK293 cells overexpressing SETD6 (Levy *et al*, 2011b). In summary, while the proteome-wide characterization of lysine methylation has recently progressed significantly, the success rates of linking a genuine methylation site to a proper biological cue have remained relatively low. However, even with the shortcomings of current methods, efforts from several groups have highlighted the roles played by lysine methylation in a myriad of cellular processes.

## Functional roles of lysine methylation

### Methylation of histone proteins

Given their abundance and ease of preparation, histone proteins were one of the first characterized methyllysine proteins (Murray, 1964). Research efforts have subsequently mapped several

methyllysine residues on histone proteins and related those modifications to specific biological cues (Fig 3) (comprehensively reviewed in (Black *et al*, 2012; Kouzarides, 2007; Shilatifard, 2006; Smith & Shilatifard, 2010). For example, methylation of histones is associated with activity at transcription start sites (H3 K4 (Santos-Rosa *et al*, 2002)), heterochromatin formation (H3K9 (Bannister *et al*, 2001; Lachner *et al*, 2001)), X chromosome silencing and transcriptional repression (H3 K27 (Cao & Zhang, 2004; Plath, 2003)), transcriptional elongation and histone exchange in chromatin (H3K36 (Carrozza *et al*, 2005; Keogh *et al*, 2005; Li *et al*, 2007a; Venkatesh *et al*, 2012; Wagner & Carpenter, 2012)) and DNA damage response (H4 K20 (Greeson *et al*, 2008; Sanders *et al*, 2004) and H3K79 (Huyen *et al*, 2004)). Our view of this network of modifications increased in complexity with the recent observation that methylation of a lysine residue influences the deposition of the same PTM on other histone proteins (Latham & Dent, 2007). The combination of different PTMs forms patterns of modifications distributed throughout the genome, and these configurations strongly correlate with the state, cell type and gene expression profile of the cell line

studied (Heintzman *et al*, 2009; Ernst *et al*, 2011; Kharchenko *et al*, 2011; Yin *et al*, 2011).

### Methylation of the transcription apparatus

The study of histone lysine methylation paved the way for the subsequent identification of an important number of sites on other proteins involved in the regulation of transcription and translation (Table 1 and Supplementary Table S1). Among those, methylation of p53 by SET7/9 (Chuikov *et al*, 2004) was initially reported to promote the pro-apoptotic activity of the transcription factor in stimulating its acetylation by p300/CBP (Ivanov *et al*, 2007). Methylation of K370 by SMYD2 was later shown to prevent the methylation of K372 by SET7/9, thus keeping p53 in a "poised" state (Huang *et al*, 2006, 2007). Methylation of K373 and K382, by G9a (Huang *et al*, 2010) and SET8 (Shi *et al*, 2007; West *et al*, 2010), respectively, were also reported to regulate the function of p53. In the first case, the modification directly inhibits p53 pro-apoptotic activity (Huang *et al*, 2010). Methylation of K382 recruits the transcriptional suppressor L3MTL1 to block the expression of p53 target genes such

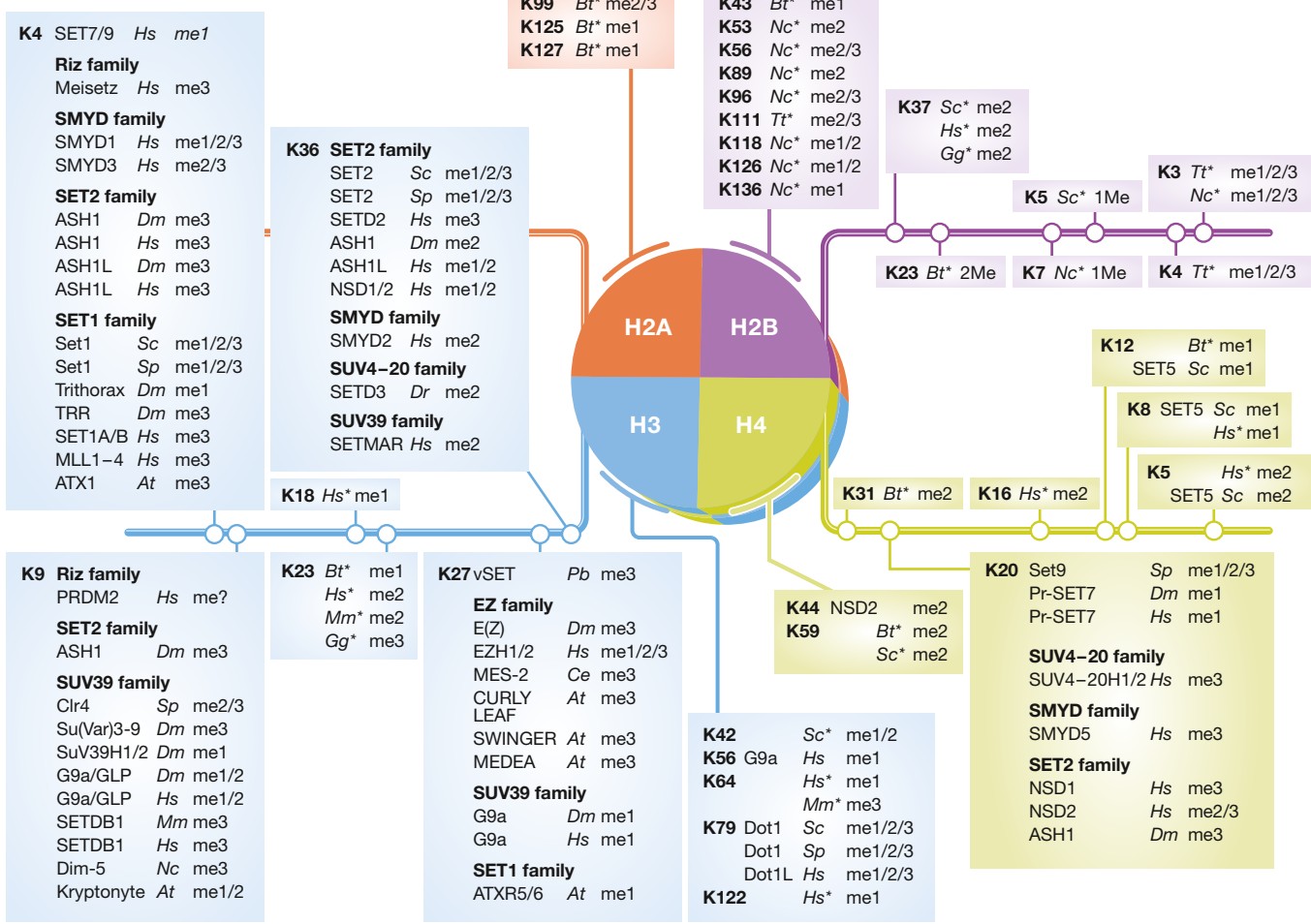

**Figure 3.   Methyllysine residues on canonical histone H2A, H2B, H3 and H4.**
Bold numbers indicate the methylated residue, italics indicate the organisms in which these modifications are found: *At, Arabidopsis thaliana*; *Bt, Bos taurus*; *Ce, Caenorhabditis elegans*; *Dm, Drosophila melanogaster*; *Dr, Danio rerio*; *Gg, Gallus gallus*; *Hs, Homo sapiens*; *Mm, Mus musculus*; *Nc, Neurospora crassa*; *Pb, Paramecium bursaria chlorella virus*; *Sc, Saccharomyces cerevisiae*; *Sp, Schizosaccharomyces pombe*; *Tt, Tetrahymena thermophila*. Known methylation states are indicated in parenthesis. A * indicates methyllysine residues modified by an unidentified enzyme.

as *p21* and *PUMA* (West *et al*, 2010). Altogether, these findings suggest that lysine methylation tunes p53 activity in a variety of ways. Intriguingly, Lenhertz *et al* (Lehnertz *et al*, 2011) and Campaner *et al* (Campaner *et al*, 2011) reported recently that SET7/9 null mice do not show any defects in p53 acetylation or apoptotic activity. However, the authors did note the possibility that other compensatory mechanisms could exist—as p53 is regulated by redundant mechanisms (Cinelli *et al*, 1998; Ryan *et al*, 2001; Kruse & Gu, 2009; Gu & Zhu, 2012; Shadfan *et al*, 2012). In addition, it remains to be investigated whether other p53 PTMs—such as the methylation of K370 by SMYD2 in control mice or redundant activating mechanisms such as the acetylation of K373 and 382—buffer the impact of SET7/9 knock-out.

Besides p53, several transcription factors are methylated by SET7/9, and as a result, their activity is modulated in different ways. Methylation of K185 inhibits E2F1 apoptotic activity by inducing its proteasomal degradation (Kontaki & Talianidis, 2010). TAF10 methylation increases its affinity for RNA polymerase II thereby stimulating the transcription of specific target genes (Kouskouti *et al*, 2004). Methylation of K630 on the androgen receptor (AR) stabilizes interaction of its N- and C-terminal domains, allowing transactivation of AR-responsive genes (Gaughan *et al*, 2011), while methylation of FOXO3 on K270 lowers the DNA binding affinity of the forkhead protein (Xie *et al*, 2012).

In apparently conflicting studies, SET7/9 was reported to methylate RelA (p65) on both K37 (Ea & Baltimore, 2009) and K314/315 (Yang *et al*, 2010b). While Ea & Baltimore (2009) showed that methylation of K37 is required for NF-κB target gene expression in HEK293 cells following TNFα stimulation, Yang *et al* (2010b) showed that, also in response to TNFα, methylation of K314 and K315 induces the proteasomal degradation of the protein in U2OS cells. It is possible that SET7/9 can methylate both residues and that another regulatory switch directs its activity specifically toward the activation or repression of RelA.

In addition to SET7/9, other methyltransferases modulate RelA activity. Methylation of K310 on p65 by SETD6 tethers GLP through its ankyrin repeat domain, promoting the deposition of the repressive mark H3K9Me2 on inflammatory response NF-κB target genes (Levy *et al*, 2011a). In contrast, cytokine stimulation induces methylation of RelA by NSD1, which promotes NF-κB activity through an unknown mechanism (Lu *et al*, 2010).

Similar to RelA, lysine methylation is a key PTM in the intricate regulatory network of the retinoblastoma protein (pRb) (Saddic *et al*, 2010; Cho *et al*, 2012a). Methylation of K810 by SMYD2 enhances pRb phosphorylation and promotes cell cycle progression, while methylation of K860 by the same PKMT stimulates the binding of the tumor suppressor to L3MBTL1 and induces cell cycle arrest (Saddic *et al*, 2010). Interestingly, following DNA damage, pRb methylation on K810 by SET7/9 leads to cell cycle arrest (Carr *et al*, 2011). Intriguingly, the same enzyme also methylates the tumor suppressor on residue K873, leading to the recruitment of HP1 to pRb target genes which also triggers cell cycle arrest (Munro *et al*, 2010).

Methyllysine residues have also been mapped on other pioneer transcription factors. Methylation of GATA4 by EZH2 regulates association of the activator to p300, regulating the expression of GATA4 target genes (He *et al*, 2012a). Similarly, methylation of C/EBP β (Pless *et al*, 2008) by G9a is important for the transactivation

potential of the transcription factor. Conversely, methylation of Reptin by the same enzyme negatively regulates a subset of hypoxia responsive genes (Lee *et al*, 2010). Taken together, these studies suggest that lysine methylation of the same residue can lead to different outcomes depending on the cellular context. Overall, it is clear that different methylation sites on the same protein can lead to drastically different effects. These findings also suggest that additional mechanisms such as feedback loops, switches and even demethylation of methyllysine residues (*see below*) will mark which lysine will be methylated during a given cellular process.

### Methylation of the translation apparatus

In contrast to the various effects reported for lysine methylation on gene transcription, investigation of the impacts of lysine methylation on translation has yielded far less details. Notably, although methylation of ribosomal proteins has been reported for three decades, the molecular and biological implications of these marks have remained elusive. Evidence that these PTMs are found in mammals, yeast, plants, bacteria and archaea lends credence to the hypothesis that methylation of the ribosome is important for its functions. However, systematic mutation of lysine residues known to be methylated failed to promote or impair either ribosomal assembly or cell survival, suggesting that methylation of ribosomal subunits plays a role in a novel, yet unexplored, biological pathway. It was recently suggested that methylation of K106 and K110 of L23ab could influence its precise positioning within the ribosome (Porras-Yakushi *et al*, 2007), while methylation of K55 on L42 might modulate association with tRNA (Shirai *et al*, 2010). However, in both cases, further experimental evidence is needed to provide a definite answer. Interestingly, recent studies have shown that the *Drosophila* Polycomb interactor Corto (centrosomal and chromosomal factor) recognizes trimethylated K3 of the ribosomal protein L12. This association, mediated by the chromodomain of Corto, recruits the RNA polymerase III and activates transcription of the heat-shock responsive gene *hsp70* (Coléno-Costes *et al*, 2012). The involvement of lysine methylation in the nuclear functions of ribosomal proteins (Bhavsar *et al*, 2010) suggests that lysine methylation of ribosome components has the potential to modulate or elicit important functions beside its canonical functions. However, given the substantial number of methyllysine residues within a ribosome (approximately 80), redundant mechanisms could mask the role of lysine methylation during translation.

## Functional diversity of lysine methylation beyond histones and transcription

In addition to transcription factors and the translation machinery, a wide variety of proteins are methylated by PKMTs, as demonstrated by both targeted and large-scale studies (Iwabata *et al*, 2005; Jung *et al*, 2008; Pang *et al*, 2010). Across all domains of life, a critical set of functions is regulated by the methylation of lysine on proteins.

### Lysine methylation & eukaryotes

Some chaperone proteins are regulated by lysine methylation in eukaryotes. For example, methylation of HSP90 by SMYD2 is involved in sarcomere assembly through titin stabilization

    

(Donlin *et al*, 2012; Voelkel *et al*, 2013). Also, SETD1 methylation of HSP70 on K561 promotes the association of the chaperone to Aurora Kinase B and stimulates the proliferation of cancer cells (Cho *et al*, 2012b). In the yeast kinetochore, methylation of Dam1 by SET1 at the yeast kinetochore is important for proper chromosome segregation during cell division (Zhang *et al*, 2005; Latham *et al*, 2011), while methylation of DNA methyltransferase DNMT1 by SET7/9 regulates global levels of DNA methylation (Estève *et al*, 2009, 2011; Zhang *et al*, 2011a). These examples demonstrate that in eukaryotes, lysine methylation is not limited to proteins of the transcriptional apparatus, but affects a wide variety of functions in the cell, many of them yet to be explored.

The role of lysine methylation in plants is even more elusive: The chloroplastic Rubisco large subunit (Houtz *et al*, 1989) and fructose 1,6-biphosphate aldolase (Magnani *et al*, 2007; Mininno *et al*, 2012) are both methylated by RLSMT, but their activity remains unaffected by the modification. Methylation of aquaporin PIP2 K3 is necessary for E6 methylation in *Arabidopsis thaliana*; yet the roles that these PTMs play remain unknown (Santoni *et al*, 2006; Sahr *et al*, 2010).

### Lysine methylation & prokaryotes

Similar to eukaryotes, lysine methylation modulates protein functions in bacteria. Methylation of pilin in *Synechocystis* sp. regulates cell motility (Kim *et al*, 2011b), while methylation of EF-Tu's K56 lowers its GTPase activity and stimulates dissociation from the membrane (Van Noort *et al*, 1986). In the latter case, levels of methylated EF-Tu increase in response to deprivation in carbon, nitrogen or phosphate levels, suggesting that extracellular cues control the activity of lysine methyltransferases (Young *et al*, 1990; Young & Bernlohr, 1991). Other lines of evidence suggest that lysine methylation of surface proteins might play a role in optimizing bacterial adherence to their environment (see *Disease implications of lysine methylation* and (Biet *et al*, 2007; Delogu *et al*, 2011; Guerrero & Locht, 2011; Soares de Lima *et al*, 2005; Temmerman *et al*, 2004)). Recent large-scale proteomic studies in *Desulfovibrans vulgaris* (Gaucher *et al*, 2008; Chhabra *et al*, 2011) and *Leishmania interrogans* (Cao *et al*, 2010) reported a large number of methylation sites on a wide variety of proteins (Supplementary Table S1), suggesting that lysine methylation is a prevalent and dynamic post-translational modification in bacteria.

### Lysine methylation & archaea

Archaea are devoid of histone proteins capable of folding DNA into octameric nucleosomes reminiscent of those found in eukaryotes. Instead, DNA compaction is achieved by a family of small basic proteins (Sandman & Reeve, 2005). As an interesting parallel with lysine methylation in eukaryotes, several of these DNA-binding proteins are methylated on lysine residues. Among those, Sac7d from *S. acidocaldarius* was the first archaeal protein reported to be methylated (Mcafee *et al*, 1995). Other members of the archaeal histone-like DNA-binding proteins, such as CCI, Cren7, Sso7c, are methylated on multiple lysine residues (Knapp *et al*, 1996; Oppermann *et al*, 1998; Guo *et al*, 2008; Botting *et al*, 2010). However, no role has yet been ascribed to this modification in the context of archaeal chromatin (Mcafee *et al*, 1996). As a possible counterpart to eukaryotes, a SET protein able to methylate the DNA-associated protein MC1-α was identified in the crenarchaea *Methanococcus*

*mazei* (Manzur & Zhou, 2005), illustrating that similar processes bring about lysine methylation across life's domains. Unique to an archaeal organism, the β-glycosidase of the hyperthermophile *Sulfolobus solfataricus* was reported to be methylated on up to five residues, a modification reported to protect the protein from thermal denaturation (Febbraio *et al*, 2004). Further proteomic studies uncovered a large number of proteins methylated in *S. solfataricus* (Botting *et al*, 2010). Interestingly, for a subset of these proteins such as the β-glycosidase, lysine methylation enhances the thermal stability of the modified protein (Fusi *et al*, 1995; Knapp *et al*, 1996; Botting *et al*, 2010). Altogether, these findings strongly suggest that lysine methylation in Archaea is equally important for proper proteome function as in Eukarya.

### Lysine methylation of viral proteins

Viruses are able to use the arsenal of methyltransferases of their host cell. Burton and Consigli, (1996) were the first to report the methylation of the major capsid protein VP1 of the murine polyomavirus. Since then, other examples of methyllysine residue have been discovered in viral proteins. Methylation of the HIV-1 transcriptional activator Tat on K50 by SETDB1 inhibits LTR transactivation (Van Duyne *et al*, 2008), while concurrent methylation of K51 by SET7/9 enhances HIV transcription (Pagans *et al*, 2011; Sakane *et al*, 2011), demonstrating that, at least in a specific context, the virus uses the host's PKMTs to ensure proper viral propagation. Some viruses also possess their own methylation machinery: *Paramecium bursara* chlorella virus 1 methyltransferase vSET site-specifically methylates histone H3 on K27 to trigger gene silencing (Manzur *et al*, 2003; Mujtaba *et al*, 2008). Overall, viruses seem to take advantage of lysine methylation mechanisms in their invasion cycle as they do of other PTMs (Gustin *et al*, 2011; Keating & Striker, 2012; Van Opdenbosch *et al*, 2012; Zheng & Yao, 2013).

## Lysine demethylation

Evidence that purified cell extracts showed slow yet detectable activity toward methylated lysine suggested that the methyl moiety added to lysine residues could be removed by dedicated lysine demethylases (KDM) (Paik & Kim, 1973, 1974). The discovery of the first KDM, LSD1, a flavine amine oxidase able to demethylate mono- and di-methylated histone H3K4, confirmed those initial reports and demonstrated that lysine methylation was part of a dynamic equilibrium. Jumonji-containing proteins, Fe(II)/α-KEG-dependent dioxygenase, were subsequently shown to demethylate tri-, di- and mono-methyllysine residues in histone proteins (Tsukada *et al*, 2006). In contrast to other KDMs, LSD1 shows a broad specificity and demethylates a large spectrum of methylated proteins. For example, demethylation of the poised K370-methylated pool of p53 by LSD1 is necessary for subsequent methylation and activation by SET7/9 (Huang *et al*, 2007). LSD1 also plays a role in the function of other transcription factors such as E2F1 (Kontaki & Talianidis, 2010), Sp1 (Chuang *et al*, 2011), STAT3 (Yang *et al*, 2010a) and MYPT1 (Cho *et al*, 2011). In addition to the demethylation of transcription factors, LSD1 also targets the DNA methyltransferases DNMT1 (Wang *et al*, 2009) and DNMT3 (Chang *et al*, 2011) and the molecular chaperone HSP90 (Abu-Farha *et al*, 2011). Notably,

demethylation of DNMT1 by LSD1 enhances its stability and regulates global levels of DNA methylation in embryonic stem cells (Wang *et al*, 2009).

Only two jumonji proteins are reported to demethylate non-histone proteins. JHDM1 (FXBL11) demethylates RelA on K218Me and K221Me, opposing the activation of this transcription factor. Interestingly, given that RelA regulates *fxbl11* gene expression, the demethylase participates in a negative feedback loop that tightly controls the activity of FXBL11 (Lu *et al*, 2010). In another study, Baba *et al* reported that the jumonji demethylase PHF2, following activation by protein kinase A, demethylates the transcription factor ARID5B. Demethylation of ARID5B stabilizes the PHF2/ARID5B complex and triggers the recruitment of PHF2's H3K9Me2 demethylase activity to, and regulate the expression of, ARID5B target genes (Baba *et al*, 2011). These examples demonstrate that demethylation is a key component of the signalization and modulation dynamics of the proteome.

## Molecular functions of lysine methylation

In comparison with other post-translational modifications, methylation appears to present only limited ways to affect the chemistry of a residue. For example, acetylation of lysine ε-amine neutralizes its positive charge and the addition of a carbonyl's dipole makes possible new types of interactions. Phosphorylation drastically modifies the charge of a protein (-3 per phosphate group) and adds a relatively important mass to an amino acid side chain (95 Da; 80 Da for Ser and Thr phosphorylation). The addition of ubiquitin and ubiquitin-like molecules, which increase the size of the targeted proteins by at least 10 kDa, is linked to cell trafficking, transcriptional regulation and endocytosis (Hicke, 2001; Haglund *et al*, 2003) and is coupled to a dedicated recognition pathway, leading to degradation by the proteasome (Glickman & Ciechanover, 2002; Ciechanover, 2005). Comparatively, methylation of a lysine residue does not modify the side chain's positive charge and causes only a small change in mass of a protein (14, 28 or 42 Da).

Following the large-scale identification of methylated lysine residues in *S. cerevisae*, Pang *et al* (2010) observed that 43% of these sites corresponded to potentially ubiquitinated residues, thus raising the possibility that methylation increases the stability of proteins by competing with ubiquitination (Fig 4A). Accordingly, pulse-chase experiments revealed an increase in the half-life of several proteins. Therefore, methylation can be considered as a regulator of ubiquitination. However, this means of regulating protein turnover rate cannot be applied to the entire proteome, as lysine methylation has been shown to increase global ubiquitination of E2F1, DNMT1, RORα and NF-κB (Estève *et al*, 2009; Yang *et al*, 2009a; Kontaki & Talianidis, 2010; Lee *et al*, 2012).

In the most direct case, methylation of a given lysine residue would preclude the addition of another modification on the same methylation site. However, "methyl switches", in which methylation of one lysine residue stimulates or inhibits the modification of at least one neighboring residue (Fig 4B), have been observed. For example, methylation of p53 K372 depends on the addition of an acetyl moiety on neighboring lysine residues (Kurash *et al*, 2008). Inhibition of cell cycle-promoting activity of E2F1 is blocked by methylation of K185, thereby stimulating the ubiquitination of the

transcription factor and preventing its phosphorylation by CK2 and ATM as well as its acetylation by PCAF (Kontaki & Talianidis, 2010). Another example is the methylation of K810 on pRb by SMYD2, which enhances phosphorylation of serine residues 807 and 811 by CDK4, inhibiting its cell cycle repressor activity (Cho *et al*, 2012a). In *S. cerevisae*, methylation of Dam1 K233 prevents the phosphorylation of S232 and S234 by Ipl1, allowing its optimal phosphorylation at S235, which promotes efficient chromosome segregation (Zhang *et al*, 2005). Overall, these observations support the fact that lysine methylation is connected to other networks of PTM and consequently to most signaling events.

In addition to controlling the deposition of neighboring PTMs, lysine methylation creates a binding surface for the recruitment of other proteins (Fig 4C). Recognition of methylated lysine residues by chromodomain proteins—part of the Royal domain family—was first reported for histone proteins (Bannister *et al*, 2001; Jacobs *et al*, 2001; Lachner *et al*, 2001; Jacobs & Khorasanizadeh, 2002). Members of the Royal domains family can specifically bind methylated lysine residues through an "aromatic cage" formed by combination of hydrophobic contacts and cation-π interactions (Ma & Dougherty, 1997; Jacobs & Khorasanizadeh, 2002; Botuyan *et al*, 2006; Hughes *et al*, 2007; Taverna *et al*, 2007). Besides the Royal family, the Plant HomeoDomain (PHD) family also reads methyl-lysine residues. Despite structural divergence between chromodomain and PHD, the methyllysine engages in similar cation-π interactions (Li *et al*, 2006; Peña *et al*, 2006; Shi *et al*, 2006; Wysocka *et al*, 2006). Interestingly, the presence of hydrogen bond networks in the aromatic cages allows the specific recognition of either mono- or di-methylated over tri-methylated lysine (Li *et al*, 2007b) triggering a specific biological response.

For instance, in histone proteins, the Polycomb complex chromodomain recognizes di- or tri-methylated H3K27 (Min *et al*, 2003), while the Eaf3 chromodomain protein recruits the Rpd3S deacetylase complex to regions enriched in H3K36 methylation (Carrozza *et al*, 2005). Among the numerous domains able to recognize methylated lysine residues on histone proteins (Musselman *et al*, 2012), the Tudor (Cui *et al*, 2012) and MBT (Kim *et al*, 2006; Li *et al*, 2007b) domains are also able to read specific methyl marks of both histone and non-histone proteins (Fig 4C). L3MBTL1 binds methyllysine residues on p53 (West *et al*, 2010) or pRb (Saddic *et al*, 2010), and the MPP8 chromodomain associates with the methylated form of DNMT3 (Chang *et al*, 2011) (Fig 4D). Interestingly, ankyrin repeats also appear to recognize methyllysine residues, as illustrated in the recruitment of GLP to methylated RelA (Levy *et al*, 2011a).

Lysine methylation can also affect biological outcomes through other mechanisms such as modulation of a protein's DNA affinity (Ito *et al*, 2007; Xie *et al*, 2011; Calnan *et al*, 2012), resistance to tryptic cleavage (Soares de Lima *et al*, 2005; Kim *et al*, 2011b) and heat denaturation (Febbraio *et al*, 2004). Overall, despite its apparently simple character, lysine methylation regulates the proteome using a wide range of mechanisms.

## Disease implications of lysine methylation

Several types of cancer involve the misregulation of PKMTs (Varier & Timmers, 2011; Butler *et al*, 2012; Greer & Shi, 2012; Hoffmann *et al*, 2012; Black & Whetstine, 2013; Campbell & Turner, 2013;

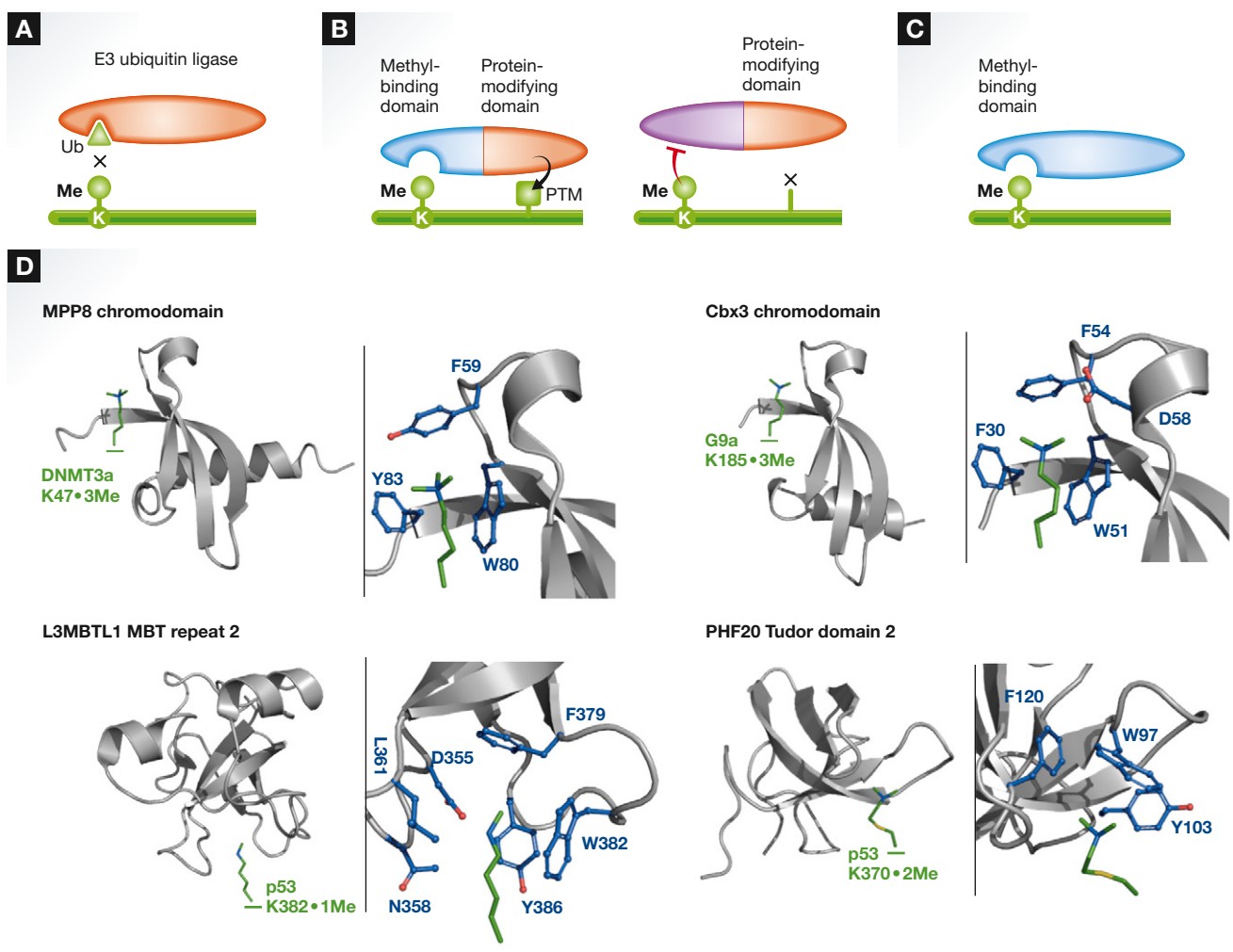

**Figure 4.  Molecular mechanisms of lysine methylation.**
(A) Lysine methylation impacts protein ubiquitination. Numerous instances of methylated lysine residues regulate protein turnover in preventing ubiquitination (see *Molecular functions of lysine methylation*). (B) Lysine methylation indirectly controls, in *cis*, deposition of other PTMs. Methyl "switches" are known to positively or negatively regulate the installation of other PTMs on neighboring residues by recruiting other protein-modifying enzymes or preventing their association with their substrates. (C) Lysine methylation controls protein-protein interactions (Examples shown in D). (D) Methyllysine residues recruit specific effector proteins. "Readers" such as the chromo, PHD finger and MBT domains can specifically bind methylated lysine residues. In addition to numerous effector proteins able to bind methylated lysine residues located on histone tails (reviewed in Musselman *et al* 2012), a few examples have been reported for non-histone substrates. In addition to HP1, the chromodomains of MPP8 and Cbx3 recognize (above) methyllysine residues (green) of non-histone proteins through residues forming an aromatic cage (blue) (PDB ID 3SVM and 3DM1). In addition, mono-methylated K382 and di-methylated K370 of p53 are bound, respectively, by the second MBT repeat of L3MBTL1 and the second Tudor domain of PHF20 (PDB ID 3OQ5 and 2LDM), respectively.

Xhemalce, 2013; Zagni *et al*, 2013). For example, expression of SMYD2 is up-regulated in esophageal squamous cell carcinoma (Komatsu *et al*, 2009) and bladder cancer cells (Cho *et al*, 2012a). SMYD3 is overexpressed in breast carcinoma and correlates with tumor proliferation (Luo *et al*, 2009), while G9a is overexpressed in hepatocellular carcinoma and contributes to lung and prostate cancer invasiveness (Kondo *et al*, 2007, 2008; Chen *et al*, 2010; Huang *et al*, 2010). Accordingly, lysine methylation has been reported to influence processes directly linked to oncogenic pathways, providing a rationale for the involvement of PKMTs in cancer. For instance, methylation of pRb by SMYD2 promotes cell proliferation, possibly through E2F transcriptional activity (Cho *et al*, 2012a). Similarly, SMYD2 methyltransferase activity prevents the activation of p53 pro-apoptotic function by the opposing modification of K372 by SET7/9 (Huang *et al*, 2006). Accordingly, these

enzymes are currently explored as efficient cancer markers and potential anti-oncogenic drug targets (Cole, 2008; Natoli *et al*, 2009; Poke *et al*, 2010; Huang *et al*, 2011; Varier & Timmers, 2011; He *et al*, 2012b; Hoffmann *et al*, 2012; Zagni *et al*, 2013).

In addition to cancer, lysine methylation plays key roles in bacterial pathogenicity. Vaccination efforts against typhus' agent *Rickettsia typhi* are targeting the immunodominant antigen OmpB. Interestingly, a critical difference between OmpB from infectious and attenuated strains is the methylation of several lysine residues of the N-terminal region of the protein (Chao *et al*, 2004, 2008). Chemical methylation of lysine residues on a recombinant peptide re-establishes serological reactivity of the OmpB fragment (Chao *et al*, 2004). In a similar fashion, *Mycobacterium tuberculosis* adhesins HBHA and LBP, important for adhesion to host cells, are also heavily methylated (Pethe *et al*, 2002; Temmerman *et al*, 2004;

Soares de Lima *et al*, 2005; Biet *et al*, 2007; Delogu *et al*, 2011; Guerrero & Locht, 2011). Similar to OmpB in *R. typhi*, immunological protection potential can be sustained by *Mycobacterium tuberculosis* HBHA only in its methylated form (Temmerman *et al*, 2004). Methylation of lysine residues in HBHA or LBP *per se* does not appear to affect the adhesive potential of the pathogen, but it instead protects the protein against proteolytic cleavage in mouse bronchoalveolar fluid, suggesting a possible role for methylation in the biology and pathogenicity of *Mycobacteria*. This hypothesis is further strengthened by the observations that the related species *Mycobacterium smegmatis* and *Mycobacterium leprae* possess methylated adhesins (Pethe *et al*, 2002; Soares de Lima *et al*, 2005). More recently, methylation of *P. aeruginosa* Ef-Tu K5 was shown to mimic the ChoP epitope of human platelet-activating factor (PAF), allowing association with PAF receptor and strongly contributing to bacterial invasion and pneumonia onset (Barbier *et al*, 2013). Given the increasing need for new and more efficient vaccines, understanding how lysine methylation impacts host–pathogen interaction will open exciting new avenues in understanding the mechanisms of pathogenicity.

## Concluding remarks

Since its discovery over half a century ago, lysine methylation has been found in all domains of life. It is a dynamic modification, as it can involve the addition of one, two or three methyl groups, and it can be reversed by dedicated demethylases. Although histone lysine methylation is held as a canonical example of the importance of this PTM, it still remains unclear whether it acts as repository of epigenetic instructions or whether it is a consequence of transcriptional and replicative DNA-based processes. Importantly, methylation of lysine residues influences protein function beyond the context of chromatin, predominantly by modulating the deposition of other PTMs such as phosphorylation, acetylation and ubiquitination or by regulating protein–protein interactions. The versatility of lysine methylation is highlighted by the fact that the same mark, mediated by different methyltransferases, can trigger distinct biological effects in different cellular contexts. Similarly, modification of different residues on a given protein by the same methyltransferase can elicit different biological responses. Future efforts involving the high-throughput analysis of protein methylation and the identification of the specific subsets of substrates attributable to each PKMT will advance our understanding of the regulatory networks underlying the lysine methylome and will provide novel functional insights regarding this PTM. Moreover, considering the involvement of protein methylation in pathologies, such analyses would be beneficial for developing diagnostic biomarkers and for revealing mechanisms of pathogenicity.

**Supplementary information** for this article is available online: http://msb.embopress.org

## Acknowledgments

Jean-François Couture acknowledges an Early Research Award from MEDI and a Canada Research Chair in Structural biology and Epigenetics. J-F C. is supported by grants from the Canadian Institutes for Health Research (GMX-209406) and the Natural Sciences and Engineering Research Council of Canada (discovery grant # 191666). Sylvain Lanouette holds a PhD Scholarship from the Fonds de Recherche en Santé du Québec (FRSQ). We would like to thank Elisa Bergamin, Pamela Zhang and William Lam for providing helpful comments on the manuscript.

## Conflict of interest

The authors declare that they have no conflict of interest.

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
