## [Review Process File · Molecular Systems Biology]

The functional diversity of protein lysine methylation

Sylvain Lanouette, Vanessa Mongeon, Daniel Figeys and Jean-François Couture

Corresponding author: Jean-François Couture, The Ottawa Institute of Systems Biology

Review timeline:

Submission date:	08 November 2013
Editorial Decision:	20 December 2013
Revision received:	31 January 2014
Editorial Decision:	11 February 2014
Revision received:	17 February 2014
Accepted:	18 February 2014

Transaction Report:

1st Editorial Decision

20 December 2013

Thank you again for submitting your work to Molecular Systems Biology. We have now heard back from the two referees who agreed to evaluate your manuscript. As you will see from the reports below, the referees acknowledge that the topic discussed in this review is relevant and timely. However, they raise a series of concerns and include suggestions for modifications, which we would ask you to carefully address in a revision of the manuscript.

Without repeating all the points listed by the referees, we would like to raise your attention to some of their more essential comments. Moreover, we would like to include a few suggestions for changes throughout the manuscript that we feel would make this review more accessible to a general audience:

- Reviewer #1 comment 1: Including more figures would indeed enhance the visibility of the key points. In addition to the existing figures, we would propose including "Figure 3" in Section 4 "Detection of Lysine Methylation" summarizing the "Methods and experimental workflows" and "Figure 4" in Section 7 summarizing the "Molecular functions of protein methylation". Please note that we will provide support from our graphics designer for the rendering and polishing of all illustrations.
- Reviewer #1 comment 1: Section 4 should be split into two parts.
- Reviewer #1 comment 3: We think that the table can be included as a Supplemental Table (as an Excel File) in its current form. Please include a column with the UniProt ID of each of the listed proteins for consistency and easier accessibility. To make the key aspects of this table accessible in a streamlined format, we would suggest including a short table (Table 1) in the main text. Table 1 should summarize key information from Table S1 i.e. using the following columns: "Protein Functions" (i.e. histones, transcription, Protein Synthesis, Chaperones etc.), "Organisms" (listing the different organisms where proteins related to each function have been found to be methylated),

"Evidence Code" (i.e. in vitro evidence, in vivo data etc.).

- Reviewer #2 comment 1: We would suggest restructuring section 5 as follows: This section could be entitled "Functional roles of Lysine Methylation" and sub-sections 5.1 and 5.2 can remain as they are. Sub-section 5.3 could be entitled "Functional diversity of lysine methylation beyond histones and transcription". In 5.3 we would recommend discussing "examples further demonstrating that K-methylation is not limited to proteins of the transcriptional apparatus but affects a wide variety of functions", listing the various functions of proteins that have been found to be methylated across life's domains (with a reference to Tables 1 and S1) etc. In the further sub-sections 5.3.1 "Examples in eukaryotes" 5.3.2 "Examples in prokaryotes" etc. the defined examples of methylated proteins can be briefly described.

- Reviewer #2 comment 2: The nomenclature should be kept consistent. Inclusion of an additional table is not necessary, but inclusion of the UniProt ID in Table S1 will be helpful.

- We think that section 2 "Uncovering Lysine Methylation" needs to be streamlined. (We would recommend reducing to half its current size.)

- Section 7: We would recommend changing the title to "Molecular functions of protein methylation". Moreover, the last part (line 543-551) can be a separate paragraph.

- Section 8: We would recommend changing the title to "Disease implications of lysine methylation".

- Please find attached a Review on PTMs that will be soon published in Molecular Systems Biology. We would like to ask you to make sure that there are no major content overlaps between the two reviews. (Please treat this manuscript as confidential information, as it has not been published online yet.)

If you feel you can satisfactorily deal with these points and those listed by the referees, you may wish to submit a revised version of your manuscript. We would like to ask you to attach a covering letter giving details of the way in which you have handled each of the points raised by the referees. Please do not hesitate to contact us if you need any clarifications regarding the above points and feel free to include additional changes as you see fit.

REFEREE REPORTS

Reviewer #1:

In this article the authors provide a historical background of lysine methylation, and review its regulatory functions in prokaryotes, archae and eukaryotes. They provide details of different approaches that are used for the identification of this PTM. The authors compiled an extensive list of lysine methylated substrates and the regulatory functions of methylation in these proteins. They also provide information about methyltransferases, demethylases and methyllysine effectors. The authors also touch upon medical relevance of methylation and list several pertinent examples of this PTM in cancer and viral infections. Throughout the manuscript they provide ample examples of different mechanisms by which lysine methylation regulates protein function.

Overall, the topic and the timing of the review are very relevant and timely. Although it is challenging to cover all the aspects of this PTM in detail, the authors have done a good job in reviewing the literature.

There are several issues with the current version of the manuscript.

1. The review covers a very broad range of topics within the lysine methylation field. The authors should carefully elaborate what is the key focus of this review. Putting a better structure and including one or two more figures could help to increase the focus and enhance the visibility the key points. For example, separating different methods of lysine methylation detection could help improve clarity. The authors could detail how methylation can be detected using radio-labeled methyl donor, site-specific methylation antibodies, and adman sequencing. Then they could focus on more contemporary, high-throughput approaches, such as a combination of mass spectrometry with heavy methyl SILAC, methyllysine-binding domain-based and antibody-based affinity enrichment approaches. Including sub-headings may help in providing clarity and distinction between these approaches.

2. Recently, Guo et al (Mol Cell Proteomics. 2013 Oct 15. [Epub ahead of print]) used an antibody-based affinity enrichment approach to identify over 1000 arginine methylation sites, and 160 lysine methylation sites. The above mentioned paper was likely published after the preparation of this review manuscript, but the authors could consider including this in the revised manuscript.

3. The table provided with the review is useful, but too long. The authors should consider dividing the table in two parts- one listing all sites with a known function and the other part where the function of modification sites is not yet known. The second part of the table could be a supplemental table, for instance.

Minor issues:

1. In the introductory paragraph (p.2, line 51-58) where the authors mention several well-known lysine PTMs they should consider mentioning ubiquitin-like lysine modifications such as SUMOylation, ISGylation and NEDDylation.

On the same note, recently lysine succinylation was mapped on a global scale (Weinert et al., Cell Rep. 2013 Aug 29;4(4):842-51; and Park et al., Mol Cell. 2013 Jun 27;50(6):919-30). If the authors wish to cite most recent and detailed studies for succinylation in this paragraph, they may consider including these citations.

2. At page 7, line 202-203, the authors should change to "Metabolic labeling methods, such as heavy methyl SILAC," this will help distinguish SILAC from in-vitro chemical labeling as well as to point out that here the authors intend to mention a different version of SILAC as opposed to regular SILAC that is based on the incorporation of heavy isotope labeled arginine and lysine.

3. At page 16, line 488-499, the authors correctly points about mass of a phosphate, but phosphorylation of serine/threonine/tyrosine adds 80Da in mass (instead of 95Da, because one of the oxygen is contributed by amino acid side chain).

4. The review contains several typos and grammatical errors, and the review would greatly benefit from a thorough proofreading.

Reviewer #2:

In this manuscript Lanouette et al. provide a comprehensive review of protein lysine methylation. The manuscript compiles all lysine-methylated proteins and their sites reported so far. Thus it will be a useful resource article for the field. There is a good description about the structural and functional aspects of this posttranslational modification. Studies on the involvement of protein lysine methylation on disease pathogenesis are also covered.

Comments:

1. The overall organization of the review is good and logical. However I think that the chapters 5.3, 5.4 and 5.5 entitled "Lysine methylation & Eukaryotes", "Lysine methylation & Prokaryotes", "Lysine methylation of viral proteins" do not belong to the main chapter 4, but could be grouped under a different heading.

2. Since this paper will be used as a resource article care should be taken to nomenclature. The authors use the standard (popular) names of the enzymes instead of the official names. It is OK, but in this case the inclusion of another table showing the respective official names and alternative names should also be included.

3. In Figure 2 (histone methylation) the same enzyme PR-SET7 is named differently for Drosophila and Humans (SETD8). This is confusing. The same name should be used.

4. The authors should check the references again. Journal names, page numbers are missing in some of them.

1- Reviewer #1 comment 1: Including more figures would indeed enhance the visibility of the key points. In addition to the existing figures, we would propose including "Figure 3" in Section 4 "Detection of Lysine Methylation" summarizing the "Methods and experimental workflows" and "Figure 4" in Section 7 summarizing the "Molecular functions of protein methylation". Please note that we will provide support from our graphics designer for the rendering and polishing of all illustrations.

We agree that more visual support would help to convey the key points of our manuscript. Figure 3, illustrating the different experimental approaches to the discovery of lysine methylation as discussed in section 4 and figure 4, illustrating the main molecular mechanisms underlying the action of lysine methylation have been prepared. In the latter case, we present the three main molecular functions of lysine methylation, including representation of the four crystal structures of methyl readers domains bound sites in a non-histone context. We limited ourselves to these in order to avoid too much overlap with other reviews on histone methylation (referenced in our manuscript). The order of paragraphs in section 8 has been modified (without altering the content) to make this clear. We would of course welcome any help you can provide to enhance the visual presentation of those figures.

2- Reviewer #1 comment 1: Section 4 should be split into two parts.

We agree that to further categorize the different approaches would help the flow of the manuscript. Section 4 was already presenting three main types of approaches in the discovery of lysine methylation in order. To best highlight those, section 4 “Detection of Lysine Methylation”, now includes three subsections”: 4.1 “Targeted discovery of lysine methylation”, 4.2, “High-throughput discovery of lysine methylation” and 4.3 “Prediction-based discovery of lysine methylation”.

3- Reviewer #1 comment 3: We think that the table can be included as a Supplemental Table (as an Excel File) in its current form. Please include a column with the UniProt ID of each of the listed proteins for consistency and easier accessibility. To make the key aspects of this table accessible in a streamlined format, we would suggest including a short table (Table 1) in the main text. Table 1 should summarize key information from Table S1 i.e. using the following columns: "Protein Functions" (i.e. histones, transcription, Protein Synthesis, Chaperones etc.), "Organisms" (listing the different organisms were proteins related to each function have been found to be methylated), "Evidence Code" (i.e. in vitro evidence, in vivo data etc.).

A systematic nomenclature would indeed greatly help to provide a more formal list of methylation sites. The UniProt ID available for each substrate has been added to the previous table 1 (now table S1), as well as an “evidence” column noting the method used to confirm the methylation site. Table 1 has been renamed “Table S1” to be placed in supplemental materials. We understand your desire to provide a more streamlined table in the body of the text that would still introduce the more complete Table S1. However, a table as described in your comments would only overlap with Figure 1 content and be sizable. As an alternative, we are now submitting a trimmed table of 2 pages instead of the initial 16 that only list the “notable” examples of lysine methylation directly referenced in our manuscript itself.

4- Reviewer #2 comment 1: We would suggest restructuring section 5 as follows: This section could be entitled "Functional roles of Lysine Methylation" and sub-sections 5.1 and 5.2 can remain as they are. Sub-section 5.3 could be entitled "Functional diversity of lysine methylation beyond histones and transcription". In 5.3 we would recommend discussing "examples further demonstrating that K-methylation is not limited to proteins of the transcriptional apparatus but affects a wide variety of functions", listing the various functions of proteins that have been found to be methylated across life's domains (with a reference to Tables 1 and S1) etc. In the further sub-sections 5.3.1 "Examples in eukaryotes" 5.3.2 "Examples in prokaryotes" etc. the defined examples of methylated proteins can be briefly described.

Section 5 is now titled “Functional Roles of Lysine Methylation”. Subsection 5.1 has been split in two: 5.1 “Methylation of the transcription apparatus” and 5.2 “Methylation of the translation

apparatus". The remainder of section 5 has been set aside as a separate section: 6 "Functional diversity of lysine methylation beyond histones and transcription".

5- Reviewer #2 comment 2: The nomenclature should be kept consistent. Inclusion of an additional table is not necessary, but inclusion of the UniProt ID in Table S1 will be helpful.

As stated above, the UniProt ID of the lysine methylation substrates has been added to table S1.

6- We think that section 2 "Uncovering Lysine Methylation" needs to be streamlined. (We would recommend reducing to half its current size.).

The size of Section 2 has been significantly reduced and now counts only 25 lines (75-100)

7- Section 7: We would recommend changing the title to "Molecular functions of protein methylation". Moreover, the last part (line 543-551) can be a separate paragraph.

The title of section 8 (previously 7) has been changed. In addition, a new paragraph now starts with "Lysine methylation can also affect..."

8- Section 8: We would recommend changing the title to "Disease implications of lysine methylation".

The title of section 9 (previously section 8) is now "Disease implications of lysine methylation".

9- Please find attached a Review on PTMs that will be soon published in Molecular Systems Biology. We would like to ask you to make sure that there are no major content overlaps between the two reviews. (Please treat this manuscript as confidential information, as it has not been published online yet.)

This manuscript was a fascinating read. However, there is no significant overlap between this manuscript and ours: lysine methylation is only succinctly alluded to.

In addition the following comments have been addressed:

Reviewer 1 comments:

10- The review covers a very broad range of topics within the lysine methylation field. The authors should carefully elaborate what is the key focus of this review. Putting a better structure and including one or two more figures could help to increase the focus and enhance the visibility the key points. For example, separating different methods of lysine methylation detection could help improve clarity. The authors could detail how methylation can be detected using radio-labeled methyl donor, site-specific methylation antibodies, and adman sequencing. Then they could focus on more contemporary, high-throughput approaches, such as a combination of mass spectrometry with heavy methyl SILAC, methyllysine-binding domain-based and antibody-based affinity enrichment approaches. Including sub-headings may help in providing clarity and distinction between these approaches.

These comments have been addressed as described in points 1 and 2 above.

11- Recently, Guo et al (Mol Cell Proteomics. 2013 Oct 15. [Epub ahead of print]) used an antibody-based affinity enrichment approach to identify over 1000 arginine methylation sites, and 160 lysine methylation sites. The above mentioned paper was likely published after the preparation of this review manuscript, but the authors could consider including this in the revised manuscript.

Indeed, this manuscript is very relevant in the context of our review: it is now included in the references cited and the section discussing the identification of lysine methylation by pan-MeK antibodies now reads: "Low specificity and sensitivity of previously available pan-methyllysine antibodies have limited the use of this approach thus far. Recently, a cocktail of antibodies was developed to enhance the enrichment of methylated peptides (Guo et al, 2013) and has successfully yielded a significant number of novel methylation sites. The new approach identified 165 sites across a wide variety of sequences in histones, elongation factors and chaperone proteins in HCT116 cells."

12- The table provided with the review is useful, but too long. The authors should consider dividing the table in two parts- one listing all sites with a known function and the other part where the function of modification sites is not yet known. The second part of the table could be a supplemental table, for instance.

Changes to the table are discussed above in point 3.

13. In the introductory paragraph (p.2, line 51-58) where the authors mention several well-known lysine PTMs they should consider mentioning ubiquitin-like lysine modifications such as SUMOylation, ISGYlation and NEDDylation. On the same note, recently lysine succinylation was mapped on a global scale (Weinert et al., Cell Rep. 2013 Aug 29;4(4):842-51; and Park et al., Mol Cell. 2013 Jun 27;50(6):919-30). If the authors wish to cite most recent and detailed studies for succinylation in this paragraph, they may consider including these citations.

The mention “ubiquitinyl and ubiquitinyl-like (SUMOylation, ISGYlation and NEDDylation)” (line 62) has been added with the relevant reference. The more recent references to lysine succinylation have been added to the text (line 64).

14- At page 7, line 202-203, the authors should change to "Metabolic labeling methods, such as heavy methyl SILAC," this will help distinguish SILAC from in-vitro chemical labeling as well as to point out that here the authors intend to mention a different version of SILAC as opposed to regular SILAC that is based on the incorporation of heavy isotope labeled arginine and lysine.

Lines 207-210 now read: In addition, metabolic labeling methods, such as heavy methyl SILAC (Ong et al, 2004), are being developed and have been applied to the *de novo*, high-throughput discovery of chromatin-specific methylation sites (Bremang et al, 2013)

15- At page 16, line 488-499, the authors correctly points about mass of a phosphate, but phosphorylation of serine/threonine/tyrosine adds 80Da in mass (instead of 95Da, because one of the oxygen is contributed by amino acid side chain).

Lines 491-493 now read: “Phosphorylation drastically modifies the charge of a protein (-3 per phosphate group) and adds a relatively important mass to an amino acid side chain (95 Da; 80 Da for Ser and Thr phosphorylation).”

16- The review contains several typos and grammatical errors, and the review would greatly benefit from a thorough proofreading.

Excluding the authors of this manuscript, three additional persons carefully proofread the manuscript. We strongly think that all grammatical errors and the typos have been corrected.

17- The overall organization of the review is good and logical. However I think that the chapters 5.3, 5.4 and 5.5 entitled "Lysine methylation & Eukaryotes", "Lysine methylation & Prokaryotes", "Lysine methylation of viral proteins" do not belong to the main chapter 4, but could be grouped under a different heading.

See point 4 above.

Reviewer 2 comments:

18- Since this paper will be used as a resource article care should be taken to nomenclature. The authors use the standard (popular) names of the enzymes instead of the official names. It is OK, but in this case the inclusion of another table showing the respective official names and alternative names should also be included.

In order to keep the main table convenient and in line with comments above, we added only the UniProt ID for each substrate (see point 3).

19- In Figure 2 (histone methylation) the same enzyme PR-SET7 is named differently for *Drosophila*

and Humans (SETD8). This is confusing. The same name should be used.

In response to the reviewer's comment on enzyme nomenclature content in Figure 2, PR-SET7 is used for both the Drosophila and Human enzyme. The figure was re-examined to assure consistency throughout.

20- The authors should check the references again. Journal names, page numbers are missing in some of them.

The references have been corrected.

2nd Editorial Decision

11 February 2014

Thank you for sending us your revised manuscript. We think that the points raised by the reviewers have been satisfactorily addressed. The manuscript has been improved in terms of structure and clarity and it nicely summarizes the current knowledge on protein lysine methylation.

Before formally accepting the manuscript, we would like to suggest a few minor changes:

- We have included some suggestions for modifications, which you will find in the attached edited version of the article. Of course these are only suggested changes and therefore please feel free to amend as you see fit.
- Figure 3 has been changed to Figure 2 and vice versa, so that the figures are numbered in the order that they appear in the text.
- We would like to ask our graphics designer to include an additional panel in Figure 4, depicting the role of lysine methylation in controlling protein-protein interactions. (This would be panel 4C in the final version of the figure and the examples that you provide -current 4C- would then be panel 4D.)

We would kindly like to ask you to let us know as soon as possible whether you agree with the modifications suggested above, so that we proceed with sending your manuscript to production in a timely manner.

2nd Revision - authors' response

17 February 2014

- We have included some suggestions for modifications, which you will find in the attached edited version of the article. Of course these are only suggested changes and therefore please feel free to amend as you see fit.

These modifications have been accepted and we will go with the title that you have suggested.

- Figure 3 has been changed to Figure 2 and vice versa, so that the figures are numbered in the order that they appear in the text.

This has been duly noted and we changed the figure numbers accordingly.